# Determination of residue levels of rodenticide in rodent livers offered novel diphacinone baits by liquid chromatography-tandem mass spectrometry

David A. Goldade[1]*, Shane R. Siers[2], Steven C. Hess[3], Robert T. Sugihara[3], Craig A. Riekena[4]

1 USDA/APHIS/WS National Wildlife Research Center, Fort Collins, CO, United States of America, 2 USDA/APHIS/WS National Wildlife Research Center, Barrigada, GU, United States of America, 3 USDA/APHIS/WS National Wildlife Research Center, Hilo, HI, United States of America, 4 Bell Laboratories, Inc., 6551 North Towne Road, Windsor, WI, United States of America

☯ These authors contributed equally to this work.
* David.A.Goldade@aphis.usda.gov

**Data Availability Statement:** The data is available at: https://doi.org/10.2737/NWRC-RDS-2023-002.

## Abstract

A specific and sensitive liquid chromatography-tandem mass spectrometry method was developed and validated for the determination of the anticoagulant rodenticide diphacinone (DPN) in mouse and rat liver. Tissue samples were extracted with a mixture of water and acetonitrile containing ammonium hydroxide. The extracted sample was cleaned up with a combination of liquid-liquid partitioning and dispersive solid phase extraction. Chromatographic separation was achieved using a Waters X-Bridge BEH C-18 LC column (50 mm, 2.1 mm ID, 2.5 μm particle size) with detection on a triple quadrupole mass spectrometer in multiple reaction monitoring (MRM) mode. The monitored transition for DPN was m/z $339.0 \rightarrow 167.0$ for quantitation and $339.0 \rightarrow 172.0$ and $339.0 \rightarrow 116.0$ for confirmation. The linear range was 0.5 to 375 ng/mL. The average precision of DPN, represented by the relative standard deviation of the observed concentrations, was 7.2% (range = 0.97% - 20.4%) and the average accuracy, represented by the relative error, was 5.8% (range = 1.06% - 14.7%). The recovery of DPN fortified at 3 different levels averaged 106% in rat liver and 101% in mouse liver. The established method was successfully used to determine DPN residue levels in Polynesian rats (*Rattus exulans*) and mice (*Mus musculus*) fed two different formulated baits containing DPN. The observed residue levels were consistent with values observed in other rodent studies. However, the amount of bait consumed was lower for the novel baits evaluated in this study.

## Introduction

Invasive rodents have been linked to extensive damage to delicate island ecosystems [1]. With no natural predators, populations often grow unchecked, impacting the ability of native flora and fauna to thrive [2,3]. Significant efforts have been undertaken to eradicate these invasive

**Funding:** This research was supported by the intramural research program of the U.S. Department of Agriculture, Animal and Plant Health Inspection Service and funding provided by Bell Laboratories, Inc. The funders took part in the study design, sample collection, decision to publish, and preparation of the manuscript.

**Competing interests:** The authors have no competing interests to the publication of this work.

rodents. These efforts often involve broadcast application of large quantities of anticoagulant rodenticide baits. Baits containing either brodifacoum or diphacinone have been used in these projects. Each of these rodenticides has the potential for unintended consequences from their use which often necessitate extensive testing to determine their safety.

Brodifacoum, a second-generation anticoagulant rodenticide, is highly efficacious for the lethal control of rodents. Brodifacoum has a drawback which makes it less attractive for island eradication efforts: it has a longer persistence in the environment than first-generation anticoagulants, such as diphacinone [4]. This difference in environmental half-life makes diphacinone more attractive for eradication efforts when large quantities will be applied in a short span of time, especially in ecologically sensitive areas or where there may be pathways to human exposure [5].

Some evidence suggests that eradications in tropical ecosystems are more prone to eradication failure [6]. For eradications to be successful, the applied baits must be both highly efficacious and palatable to the rodents. To address potential palatability issues, novel bait matrices under development for deployment in eradication operations were formulated at 50 and 100 ppm diphacinone and offered to wild-caught Polynesian rats (*Rattus exulans*) and house mice (*Mus musculus*) in a two-choice test with the test animals being offered both the formulated bait and a standard maintenance diet. The livers were harvested from the mice and rats and analyzed for diphacinone residues by liquid chromatography with tandem mass spectrometry to verify bait consumption and inform potential risk assessments from secondary exposures.

Rodenticides have been quantified in a variety of matrices by many different analytical techniques. Liquid chromatography paired with ultra-violet or fluorescence detection was the industry standard for many years. This technique was able to achieve detection limits in the low parts-per-million level in most cases. In more recent years advances in tandem mass spectrometry has resulted in its emergence as the preferred technique for low level determinations of rodenticide residues [7]. Combining both selectivity and sensitivity improvements, LC-MS-MS produces superior performance often achieving detection in the sub part-per-billion range [7]. Due to the enhanced selectivity afforded by tandem mass spectrometry, minimal sample clean-up is often needed to permit the detection of rodenticides in complex biological matrices [8–10]. The method discussed in this work takes advantage of these advancements in sample preparation and analysis to determine the residue levels of diphacinone in field-captured rodents. These results will be used to inform potential decisions on the safety and efficacy of the novel baits being developed for the control of invasive rodents.

## Materials and methods

### Trapping of rats and mice

Ninety-two Polynesian rats (*Rattus exulans*) and ninety-two house mice (*Mus musculus*) of mixed genders were trapped from wild populations in Hawaii with no known exposure to anticoagulant rodenticides and transported to the National Wildlife Research Center's Hawaii Field Station. The rodents were held in quarantine for not less than 7 days, during which they were maintained in individual housing and given free access to water and a maintenance diet (LabDiet™ Laboratory Rodent Diet 5001, Lab Diet, St. Louis, MO, USA).

### Bait testing

The rats and mice were randomly divided into test and control groups. On test day one half of the test animals were offered either a 50 or 100 ppm DPN formulated bait comprised of compressed cereal pellets with a proprietary formulation (Bell Laboratories, Madison, WI, USA).

The test animals were also offered a USEPA standard challenge diet [11] in a two-choice test. Control animals received only the standard animal feed. The rodents were allowed to feed *ad libitum* for 15 days, followed by a 10-day holding period during which the formulated baits were removed, and all animals received only the standard animal feed diet. Consumption of bait and challenge diet (g) was recorded daily. Twice daily, the condition of the rodents was observed and noted. Deceased animals were removed; their weight and the time were recorded. No individuals from the control group expired during the exposure period. All rodents that survived the full 15-day exposure period and 10-day post-exposure period were euthanized with $CO_2$. All animals were euthanized humanely in accordance with American Veterinary Medical Association [12] standards and practices.

**Justification for use of study animals.** The purpose of this research was to evaluate a new bait formulation for the lethal control of invasive rodents in fragile island ecosystems. Due to potential biological variability within a population as well as potential gender-dependent metabolic difference, the use of *in*-vivo testing was the only suitable approach for a study of this type. The potential risk of a failed eradication necessitated the inclusion of death as an endpoint for the study animals. The use of sedatives, analgesics, or antidotes during the study was rejected on the following basis: DPN is an anti-coagulant which is detoxified in the liver [13]. Any veterinary drug which could enhance or suppress specific metabolic pathways could lead to a false assessment of the efficacy of the bait under evaluation. The number of animals selected were the minimum required to produce adequate efficacy data.

**Humane treatment of test animals.** All procedures involving animals were carried out with the approval of the NWRC Animal Care and Use Committee (NWRC protocol QA-2546) and in accordance with Good Laboratory Practices (40 CFR Part 160). Despite the need for study animal mortality, significant efforts were made to minimize suffering. Three criteria were established to determine if an animal should be humanely euthanized before the end of the study period. Firstly, the rodent must have consumed bait sufficient to meet the LD100 value for that species. Secondly, the animals were observed twice daily by trained animal care staff as verified by the NWRC Animal Care and Use Committee. If during that observation period the animals displayed any of the following clinical signs, they were immediately euthanized: A body condition score of -2 as defined by Foltz and Ullman-Cullere [14], difficulty breathing or rapid breathing over two successive observations, lateral recumbence over two successive observations, weight loss of greater than 10%, or blood in the feces or urine. Lastly, any animal exhibiting agonal vocalizations or persistent convulsions was immediately euthanized, independent of other criteria. Under the conditions listed above, no study animals required euthanasia prior to the end of the study period.

## Tissue collection

All rodents were necropsied to remove the liver for analysis. An incision was made in the skin covering the abdomen and the skin was pulled back. A lateral incision was then made at the base of the breastbone and a pair of scissors used to cut the breastbone on each side. The liver was removed from each rodent and ground to a fine powder using a liquid nitrogen freezer mill (SPEX CertiPrep, Metuchen, NJ, USA) and stored in individual vacuum sealed bags at -30˚C until analysis.

## Liquid chromatography-mass spectrometry analysis

Water (200 μL) was added to the homogenized liver samples (0.075 to 0.125 g) in a 15-mL polypropylene centrifuge tube and the sample vortexed to produce a slurry. The samples were spiked with 50 μL of surrogate dissolved in acetonitrile (20.0 μg/mL d4-diphacinone; CDN

**Table 1. Liquid chromatograph/mass spectrometer conditions.**

| Injection Volume: | 5 μL | | | | | |
|---|---|---|---|---|---|---|
| Column Temperature: | 40°C | | | | | |
| Column Flow: | 0.350 mL/minute | | | | | |
| Mobile Phase A: | 80% 10-mM ammonium acetate buffer/20% acetonitrile | | | | | |
| Mobile Phase B: | Acetonitrile | | | | | |
| | Gradient Program: | | | | | |
| | Time (min) | Percent A | Percent B | | | |
| | 0.00 | 100 | 0 | | | |
| | 0.50 | 100 | 0 | | | |
| | 5.00 | 40 | 60 | | | |
| | 5.10 | 0 | 100 | | | |
| | Run time = 5.50 minutes | | | | | |
| | Post-run equilibration time = 0.75 minutes | | | | | |
| Ion Source: | Agilent Jet-Stream ESI Negative | | | | | |
| Gas Temperature: | 225°C | | | | | |
| Gas Flow: | 6 L/minute | | | | | |
| Nebulizer: | 25 psi | | | | | |
| Sheath Gas Temperature: | 375°C | | | | | |
| Sheath Gas Flow: | 12 L/minute | | | | | |
| Nozzle Voltage: | 0 V | | | | | |
| Compound | Precursor Ion (m/z) | Product Ion (m/z) | Collision Energy (V) | Fragmentor (V) | Dwell (ms) | |
| d4-Diphacinone | 343.0 | 167.0 | 26 | 120 | 100 | |
| Diphacinone | 339.0 | 167.0 | 26 | 120 | 100 | |
| Diphacinone | 339.0 | 172.0 | 20 | 120 | 100 | |
| Diphacinone | 339.0 | 116.0 | 50 | 120 | 100 | |

Isotopes, Pointe-Claire, Quebec, Canada) and extracted with 5 mL of 1% ammonium hydroxide in acetonitrile (Fisher Scientific, Waltham, MA, USA). A solvent/water partition was created by the addition of 200 mg of a Quechers salt packet (Agilent Technologies, Santa Clara, CA, USA). An aliquot (1.5 mL) of the acetonitrile layer was transferred to a dispersive solid phase extraction tube (Agilent Technologies, Santa Clara, CA, USA) containing 25 mg C18 and 150 mg magnesium sulfate. A 1-mL aliquot of the sample was evaporated to dryness in a vacuum concentrator (Vacufuge; Eppendorf, Enfield, CT, USA) and reconstituted in 80% 10-mM ammonium acetate:20% acetonitrile. Analysis was performed using an Agilent 1290 Liquid Chromatograph attached to an Agilent 6470A QQQ Mass Spectrometer (Santa Clara, CA, USA). Separation was achieved using a 50 mm x 2.1 mm ID, 2.5 μm, X-Bridge BEH C18 column (Waters Corporation, Milford, MA, USA). Other instrument parameters are listed in Table 1.

## Method validation

The method was validated by evaluating the following parameters: selectivity, limits of detection (DL) and quantification (QL), linearity, accuracy, and recovery. Unless otherwise specified, result values were tabulated and statistical analysis was performed using Microsoft Excel 15.0 (Microsoft Corp, Redmond, WA, USA).

**Control tissues.** The repeatability and performance of the method was assessed using Polynesian rats (*Rattus exulans*) and house mice (*Mus musculus*) trapped at the same time as

the study animals. These rodents received only the standard maintenance diet for the duration of the study. Their tissues were processed according to the procedures outlined above.

**Selectivity.**   Selectivity of the method was evaluated by analyzing 11 replicate samples of liver tissue from wild-caught Polynesian rats and 8 replicate samples of liver tissue from wild-caught house mice. Potential interferences were assessed at the retention time of DPN by comparing the analytical response in the control samples to samples fortified near the QL.

**Detection of quantification limits.**   The detection limit (DL) for DPN in Polynesian rat and house mouse liver was estimated from the mean chromatographic response of control samples compared to control samples that had been fortified to 20 ng/g with DPN. The DL was defined as the concentration of DPN required to generate a signal equal to 3X the baseline noise (measured peak-to-peak) observed in the baseline at the retention time of DPN in the control samples. The quantitation limit (QL) for DPN in Polynesian rats and house mice liver was estimated in a similar fashion to the DL with the multiplier of 10X baseline noise used instead of 3X.

**Linearity.**   Two sets of eight calibration standard solutions were prepared ranging from 0.470 ng/mL to 375 ng/mL (corresponding to a nominal concentration in liver tissue samples of 23.5 ng/g to 18,750 ng/g) by serial dilution into a solution of 10-mL ammonium acetate in water. Each liver tissue sample of 100 mg results in a final extract of 5 mL which produces a 1:50 dilution factor in the sample results. Each standard was injected in duplicate; the response ratio of DPN area to d4-DPN area was plotted against relative concentration. To improve accuracy of responses at low calibration levels, a weighted (1/x) quadratic regression was performed on the data set.

The accuracy of the calibration standards was determined for each data set by calculating the observed concentration of each standard in the calibration curve using the regression equation. This calculated observed concentration was then compared to the theoretical concentration of the standard to determine the accuracy as a percentage.

**Recovery.**   A minimum of 8 replicates at each fortification level were prepared by adding an appropriate aliquot of a fortification solution prepared in acetonitrile to a 100 mg sample of control liver tissue. Control Polynesian rat and house mouse livers were fortified at nominal concentrations of 15.7, 313, and 18,800 ng/g.

The accuracy of the method was determined by calculating the relative error of the mean observed concentration at each fortification level. This value was calculated by determining the absolute value of the difference between observed concentration and target concentration and dividing by the target concentration.

**Statistical methods.**   The statistical model used to evaluate the variance of the quality control data was SAS's General Linear Model (GLM; SAS Institute, Inc., Cary, NC). The GLM performs analysis of variance by using least squares regression to fit general linear models. The response was the observed residue level in the liver samples. The fixed effects were species type, gender, concentration of the formulated bait, number of days the rodent survived the test, and the amount of bait consumed. The interaction between fixed effects was also evaluated.

## Results and discussion

### Method development

A deuterated form of DPN (d4-DPN) was selected as the internal standard (IS) for this method. Multiple reaction monitoring (MRM) transition pairs were selected for both DPN and d4-DPN and optimized with respect to fragmentation and collision energy using Masshunter's Method Optimizer software (Agilent Technologies, Santa Clara, CA, USA). The

quantitative transition for DPN was m/z 339.0→167.0 while the d4-DPN transition was m/z 343.0→167.0.

Initial experiments were conducted using a dSPE cartridge containing primary-secondary amine (PSA) in addition to the C-18 and magnesium sulfate in the validated method. The PSA-containing dSPE tubes resulted in lower overall recovery of both the DPN and d4-DPN. The absolute (not corrected for IS response) recovery from the PSA-containing dSPE cartridge averaged approximately 40%, while that for the dSPE cartridge containing only C-18 and magnesium sulfate was approximately 85%.

A small amount of base (ammonium hydroxide) was added to the extraction solvent to improve clean-up of the sample extract. Significant matrix co-extractants were present when acetonitrile was used without the ammonium hydroxide present.

## Method validation

**Selectivity.** Typical MRM chromatograms of unfortified control tissues, control tissues fortified at 15.7 ng/g for Polynesian rat liver, and from a Polynesian rat which fed on a 50 ppm DPN bait formulation are presented in Figs 1–3. Matrix peaks were observed near the retention time of DPN in most of the matrices tested (Fig 1). These peaks occurred after the retention time for DPN and did not interfere with the accurate detection of DPN. Under the conditions specified, the retention times of d4-DPN and DPN were 2.6 minutes.

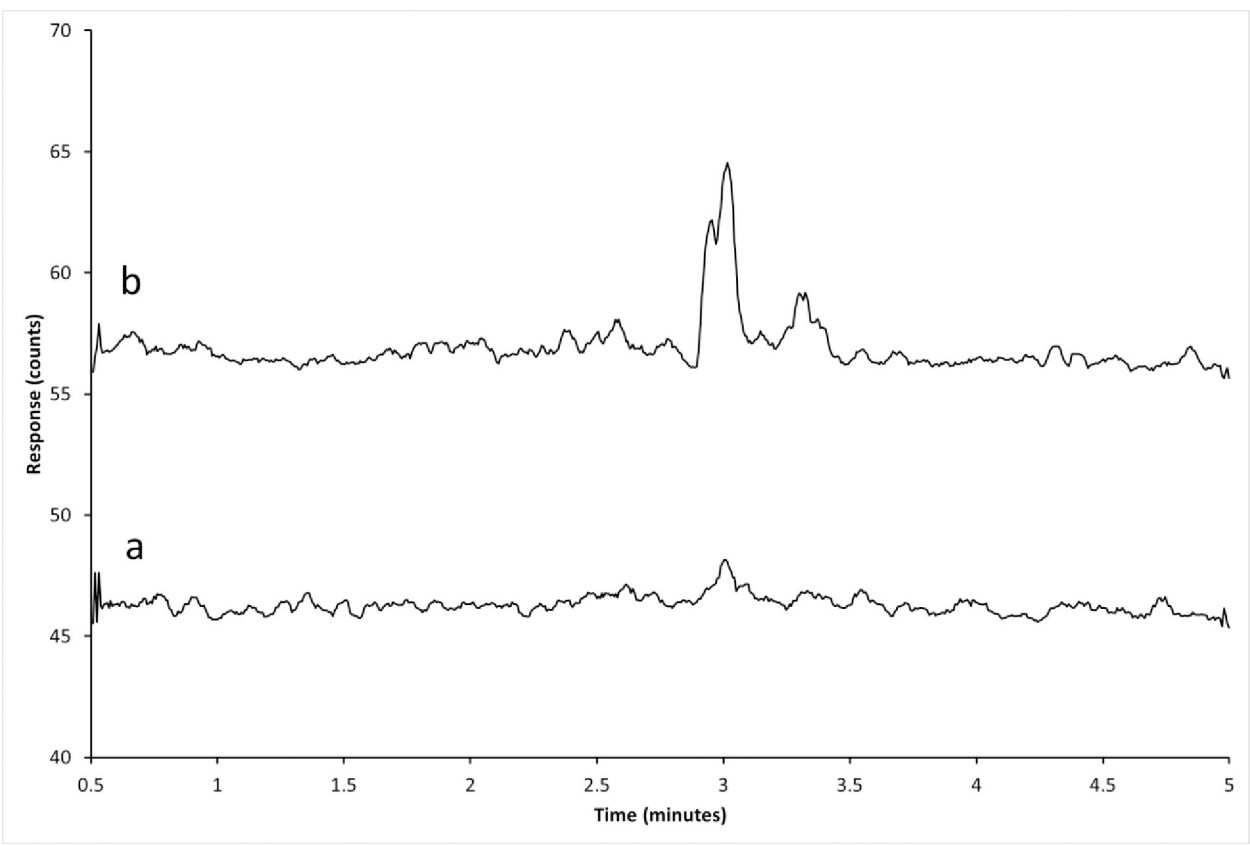

**Fig 1. Chromatograms of liver tissue from an untreated sample.** Liver tissue extracts from Polynesian rat (a) and house mouse (b). The *m/z* 339.0 → 167.0 MRM transition is shown. The approximate retention time of DPN is 2.6 minutes.

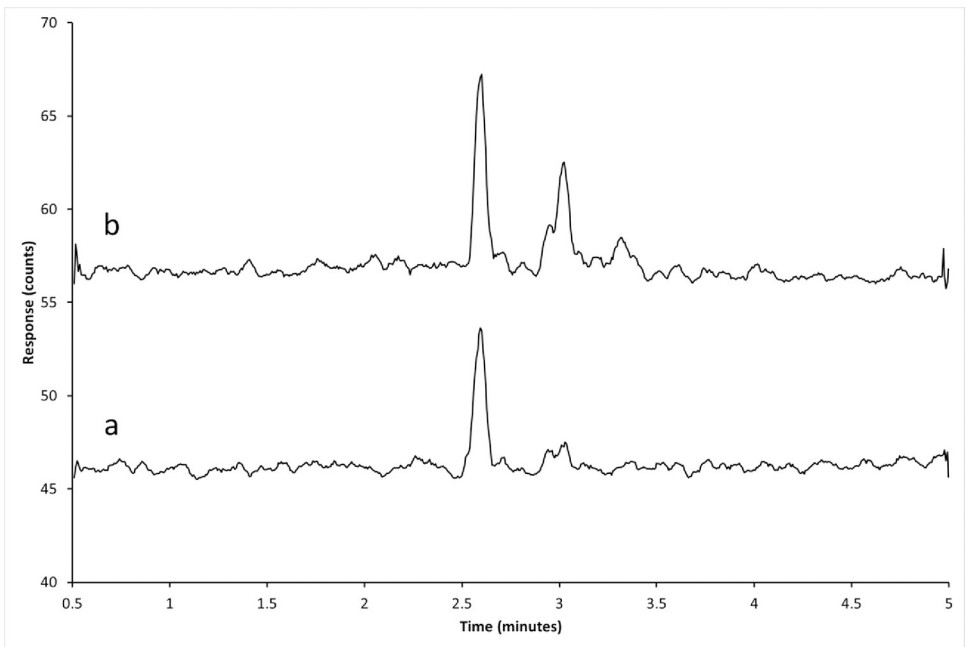

**Fig 2. Chromatograms of liver tissue from an untreated sample which were spiked with DPN at 15.7 ng/g.** Spiked liver extracts of control liver tissue from (a) Polynesian rat (b) house mouse. The $m/z$ 339.0 → 167.0 MRM transition is shown. The approximate retention time of DPN is 2.6 minutes.

**Detection and quantification limits.** The detection limit (DL) for DPN in rodent livers was estimated from the mean chromatographic response of control samples compared to control samples that had been fortified to 15.7 ng/g with DPN. The DL and QL for Polynesian rat and house mouse liver tissue are shown in Table 2.

**Linearity.** The average of all calibration curves was determined (n = 9) where Y represents the ratio of concentration of DPN to d4-DPN and X represents the ratio of DPN area response to d4-DPN area response (Table 3). Calibration was achieved with a 1/x weighted quadratic regression analysis ($Y = aX^2 + bX + c$) with correlation coefficients of 0.999 (0.99993 ± 0.000033) or better (Fig 4). The calibration curve was linear over the range of 0.47 ng/mL to 375 ng/mL DPN. The accuracy of all concentrations in the calibration curve was <10%.

**Precision and accuracy.** The precision and accuracy values for recovery of DPN from rodent livers demonstrated good method performance across all levels (Table 4). The samples were analyzed over the course of a 1-month period. The average precision of DPN, represented by the relative standard deviation of the observed concentrations, was 7.2% (range = 0.97%– 20.4%) and the average accuracy, represented by the relative error, was 5.8% (range = 1.06% - 14.7%). Therefore, the validated method provided acceptable precision and accuracy for the determination of DPN in liver tissues.

ANOVA results from a GLM analysis (SAS General Linear Model (GLM); SAS Institute, Inc., Cary, NC) demonstrated a significant effect from species when it was included in the model (F = 5.64; p = 0.0214), therefore the recovery results from the two species were examined separately. For the rat samples, fortification level was a significant factor (F = 37.53; p = <0.0001) with low, mid, and high levels all being significantly different from each other (α = 0.05). For the mouse samples, there was no significant difference between fortification levels (F = 0.84; p = 0.4561).

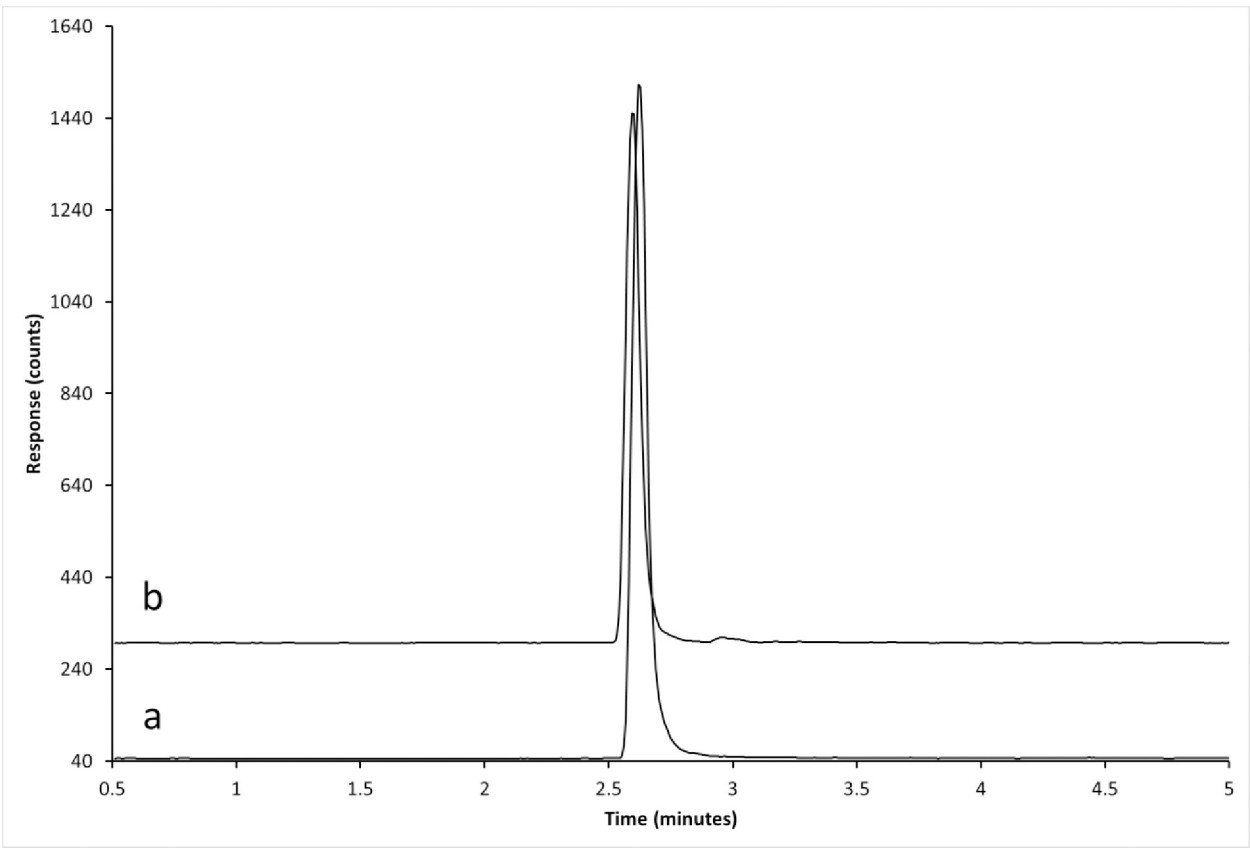

**Fig 3. Chromatograms of liver tissue from rodents offered a formulated bait containing 50 ppm DPN.** Liver tissue extracts from (a) Polynesian rat (b) house mouse which had fed on the 50 ppm formulated bait. The $m/z$ 339.0 → 167.0 MRM transition is shown. The approximate retention time of DPN is 2.6 minutes.

**Rodent feeding study.** Statistical analysis was performed on the data set using ANOVA and a GLM analysis (SAS General Linear Model (GLM); SAS Institute, Inc., Cary, NC). The results demonstrated a significant effect when species was included in the model (F = 20.98; p = <0.0001), therefore further analysis was performed on each subset of data independent of the other. When testing for the effect of rodent gender on the data sets, the results were significant for rats (F = 8.09; p = 0.0066) but not for mice (F = 3.36; p = 0.0732). Bait formulation level had a significant effect for mice (F = 6.15; p = 0.0167) but not for rats (F = 2.35; p = 0.1319). When considering the impact of the amount of treated bait consumed and the number of days the rodent fed on the bait, the concentration of the formulated bait (F = 11.42; p = 0.0016), the interaction of amount of bait consumed and formulation level (F = 9.13; p = 0.0042), and the interaction of all three terms were significant with respect to mice (F = 3.91; p = 0.0543). For rats only the concentration of the formulated bait had a significant effect on residue level (F = 4.92; p = 0.0320). These results, taken as a whole, indicate that the

**Table 2. Detection and quantitation limits for rodent liver tissue analyzed using LC/MS/MS.**

| Species | Detection Limit | Quantitation Limit |
|---|---|---|
| Polynesian Rat | 3.5 ng/g | 11.5 ng/g |
| House Mouse | 4.8 ng/g | 15.9 ng/g |

**Table 3. Average regression coefficients (n = 9).**

| Coefficient | Mean |
|---|---|
| a | 0.0083 ± 0.0081 |
| b | 1.07 ± 0.017 |
| c | 0.00024 ± 0.00012 |

formulated product had less impact on the observed residue level in rodent liver for the rats than it did the mice in this study.

Both formulation levels proved highly effective at producing mortality in Polynesian rats with all animals succumbing to the rodenticide in the 50 ppm group (Table 5) and 25 of the 26 animals perishing in the 100 ppm group (Table 5). Efficacy was much lower in the house mouse treatment groups. The 50 ppm and 100 ppm treatment groups had 5 and 7 of the 26 mice in each group survive the entire study period, respectively (Table 5). It is possible that some taste aversion could be occurring among the mice. The mice that succumbed to the 50 ppm formulated bait had 30.4% average bait acceptance calculated as the ratio of the mass of treated bait consumed to all feed consumed during the two-choice trial. Mice that survived the entire study period only had 10.2% average acceptance. This indicates potential palatability or bait avoidance problems in the mouse treatments groups.

Another possible explanation for the difference in survival between rats and mice is the difference in toxicity for DPN in these two classes of rodents. The published LD50 for rats is 0.3 to 2.3 mg/kg [15] while that for mice is 340 mg/kg [16]. Although the mice were much smaller (average 11.5 g) than the rats (average 55.2 g) in terms of body weight, the difference in LD50 means the mice would have to consume more of the treated baits than the rats to achieve a lethal dose.

## Conclusion

A method for the quantitation of DPN in rodent liver tissues was developed and validated. The method was successfully used to quantitate residue levels of DPN in wild-caught Polynesian rats and house mice fed a formulated bait containing DPN at two levels (Table 6).

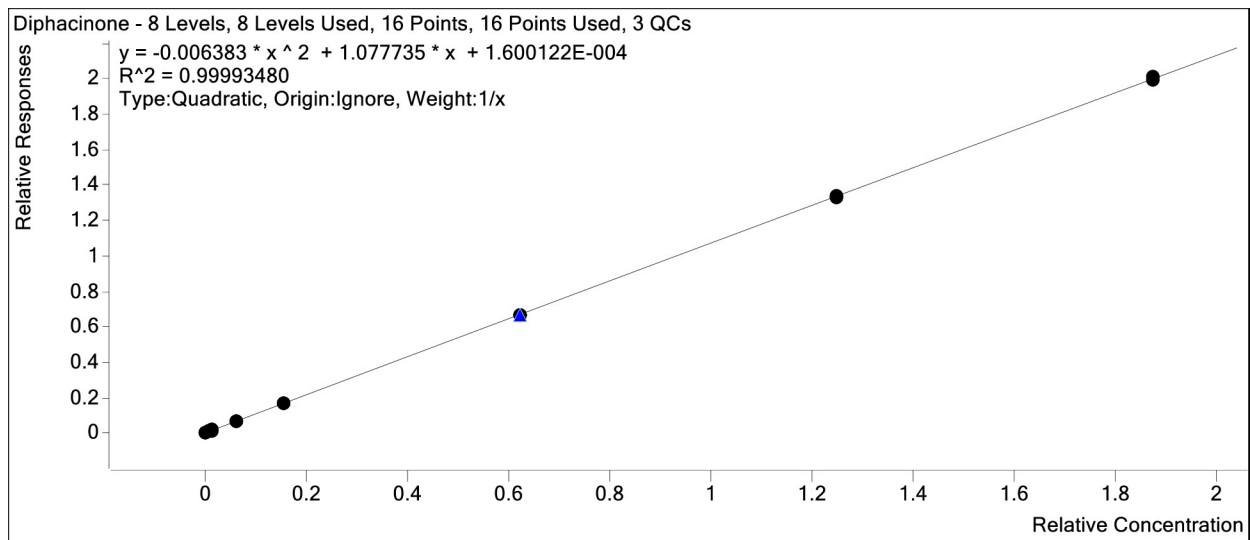

**Fig 4. Representative calibration curve for DPN.** Calibration standard solutions were prepared ranging from 0.470 ng/mL to 375 ng/mL (corresponding to a nominal concentration in liver tissue samples of 23.5 ng/g to 18,800 ng/g).

**Table 4. Precision and accuracy of DPN in rodent liver tissues.**

| Species | n | Target (ng/g) | Observed (ng/g) | Precision (RSD,%) | Accuracy (RE,%) |
|---|---|---|---|---|---|
| Polynesian Rat | 11 | 15.7 | 19.6 ± 4.0 | 20.4% | 14.7% |
| | 11 | 313 | 329 ± 12 | 3.65% | 4.72% |
| | 9 | 18800 | 19100 ± 720 | 3.77% | 1.20% |
| House Mouse | 7 | 15.7 | 18.8 ± 2.3 | 12.2% | 10.6% |
| | 8 | 313 | 321 ± 7.7 | 2.40% | 2.45% |
| | 8 | 18800 | 18600 ± 180 | 0.97% | 1.06% |

**Table 5. Mortality for two species of rodents fed a diphacinone-treated bait.**

| Species | Formulation Level | |
|---|---|---|
| | 50 ppm | 100 ppm |
| Polynesian Rat | 26/26 (100%) | 25/26 (96.2%) |
| House Mouse | 21/26 (80.8%) | 19/26 (73.1%) |

**Table 6. Mean DPN residue levels (µg/g) in rodent livers following feeding on treated baits.**

| Formulation Level | | Polynesian Rat | House Mouse |
|---|---|---|---|
| 50 ppm | Mean ± sd = | 6.63 ± 5.48 | 2.00 ± 1.92 |
| | Range = | 0.613–19.6 | 0.196–8.28 |
| | RSD = | 83% | 96% |
| | n = | 26 | 26 |
| 100 ppm | Mean ± sd = | 8.96 ± 6.82 | 4.82 ± 5.53 |
| | Range = | 0.855–31.4 | 0.097–18.70 |
| | RSD = | 76% | 115% |
| | n = | 24 | 25 |

In a secondary hazard study of multiple rodenticides, including diphacinone, Fisher et al. [17] found a mean liver residue in rats that fed on a 50-ppm diet and succumbed to a lethal dose of 4.7 µg/g with a range of <0.1 to 9.0 µg/g. During two studies performed in Hawaii, the average liver residue observed in rats was 3.7 µg/g for a hand-baiting study [18] and 4.4 µg/g during an arial application [19]. The cited studies were conducted with 50 ppm diphacinone baits. These values compare very well with the results of the current study. While the liver residue levels are quite similar, the amount of treated bait needed to produce fatality in the rats was significantly less for this study with an average of 15.9 g of the 50 ppm bait being consumed while an average of 113 g was consumed in a similar feeding study [19]. The novel baits evaluated were efficacious in rats, produced similar residue levels to currently existing baits, and required significantly less bait to achieve similar results.

## Author Contributions

**Conceptualization:** Shane R. Siers, Craig A. Riekena.

**Data curation:** David A. Goldade.

**Formal analysis:** David A. Goldade.

**Funding acquisition:** Craig A. Riekena.

**Investigation:** Shane R. Siers, Robert T. Sugihara.

**Methodology:** David A. Goldade, Shane R. Siers.

**Project administration:** Shane R. Siers, Steven C. Hess, Craig A. Riekena.

**Resources:** Shane R. Siers, Craig A. Riekena.

**Supervision:** Shane R. Siers, Steven C. Hess.

**Writing – original draft:** David A. Goldade.

**Writing – review & editing:** Shane R. Siers, Steven C. Hess, Robert T. Sugihara, Craig A. Riekena.

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
