## [Decision Letter · Decision Letter 0]

2 May 2023

PONE-D-23-07599Determination of residue levels of rodenticide in rodent livers offered novel diphacinone baits by liquid chromatography-tandem mass spectrometryPLOS ONE

Dear Dr. Goldade,

Thank you for submitting your manuscript to PLOS ONE. After careful consideration, we feel that it has merit but does not fully meet PLOS ONE’s publication criteria as it currently stands. Therefore, we invite you to submit a revised version of the manuscript that addresses the points raised during the review process.

We look forward to receiving your revised manuscript.

Kind regards,

Totan Adak

Academic Editor

PLOS ONE

Journal Requirements:

Additional Editor Comments:

Prepare the tables as per the journal format.

Authors should cite the relevant references on Bait testing, Justification for use of study animals, Humane treatment of test animals

Reviewers' comments:

Reviewer's Responses to Questions

**Comments to the Author**

1. Is the manuscript technically sound, and do the data support the conclusions?

Reviewer #1: Partly

Reviewer #2: Yes

2. Has the statistical analysis been performed appropriately and rigorously? 

Reviewer #1: Yes

Reviewer #2: No

3. Have the authors made all data underlying the findings in their manuscript fully available?

Reviewer #1: No

Reviewer #2: Yes

4. Is the manuscript presented in an intelligible fashion and written in standard English?

Reviewer #1: Yes

Reviewer #2: Yes

5. Review Comments to the Author

Reviewer #1: Line 30.. “The sample was cleaned up with a combination….” Should be stated as “The extracted sample was cleaned up with a combination…”

Line 31.. “Separation was achieved….” Should be stated as “chromatographic separation was achieved…”

Line 35 and 37... State precision and accuracy in range instead of average.

Line 73,74, 77, References were cited in form of numbers as they appeared in the document, but some reference were cited with first author name only. Arrange the manuscript as per journal instructions.

Line 90. “On test day 0 half of the test animals were offered…” should be “On test day one half of the test animals were offered…”

Line 100-102. All animals were euthanized humanely in accordance with American Veterinary Medical Association standards and practices. Cite reference to this statement.

Line 110. DPN is an anticoagulant which is detoxified in the liver. Cite reference to this statement.

Line 138-139. ‘The samples were fortified with 1 μg of surrogate dissolved in acetonitrile” should be stated as The samples were spiked with ____µl of surrogate dissolved in acetonitrile (µg/L).

Line 154. accuracy, recovery. Should be stated as accuracy and recovery.

Line 154-155. What does this statement mean “When not specified”, result values were tabulated and analyzed using Microsoft Excel 15.0 (Microsoft Corp, Redmond, WA,USA). This statement seems irrelevant.

Line 175. Coefficient of determination was not mentioned. Without coefficient of determination, linearity is meaningless.

Line 176. Authors are needed to explain how does 0.470 ng/mL to 375 ng/mL is corresponding to a nominal sample concentration of 4.7 ng/g to 3750 ng/g. Authors have mentioned linearity in ng/mL but DL and QL in ng/g. Is there any specific reason to do this? Similarity authors also used positive controls 20, 350 and 12500 ng/g and linearity range 0.470 ng/mL to 375 ng/mL. Authors are needed to remove this ambiguity.

Line 204. Authors also required to discuss in detail experiments performed during method development regarding instrumental parameters such mobile phase, gradient elution, ion source parameters etc.

Line 229-232. The DL was defined as the concentration of DPN required to generate a

signal equal to 3X the baseline noise (measured peak-to-peak) observed in the baseline at the retention time of DPN in the control samples. The quantitation limit (QL) for DPN in rodent livers was estimated in a similar fashion to the DL with the multiplier 232 of 10X baseline noise used instead of 3X. These lines should be removed as these terms already defined in lines 170-174.

Line 297. a range of <0.1 tp 9.0 μg/g should be written as a range of <0.1 to 9.0 μg/g.

Additional comments.

Authors did not mention validation guidelines they have used for this study. Moreover one of the important validation parameter when ESI is used as ion source is ion suppression which was not discussed. Instrumental plotted calibration curve with coefficient of determinations should be shown in the form of figure. Table 1, the gradient program at 5.10 and 5.50 min is same and needed to be corrected.

To conclude, I would advise against accepting this manuscript for publication unless significant revisions are made.

Reviewer #2: The manuscript "Determination of residue levels of rodenticide in rodent livers offered novel diphacinone baits by liquid chromatography-tandem mass spectrometry" describes the determination of DPN residue levels in Polynesian rats (Rattus exulans) and mice (Mus musculus) fed two different formulated baits containing DPN. Some of the the sections are not well described and discussed in a very inefficient manner. I strongly recommend the major revision of the manuscript through the incorporation of the following comments although I found it is a generally interesting story.

1) Abstract, and section headings need to be revised along with more quantitative and background information. The abstract does not summarize the objectives, the principal results, and major conclusions of the present study.

2) The originality of the paper needs to be further clarified. The present form does not have sufficient results to justify the novelty of a high-quality journal paper (there is a lot of similar studies that depicted determination of rodenticide levels). The background of the study, the research gap/scientific novelty, and the state-of-the-art (and purpose) of the study are not clearly illustrated. The hypothesis and justification of the study needs to be elaborated.

3) Have you looked for the chirality of the DPN/and their stereoisomers through chiral column analysis? What about the optimization of the sample preparation process and LC-MS/MS conditions?

4) Any information about matrix-matched standards and fortified liver matrix sample preparation and evaluation of the extraction procedures?

5) What about metabolites and their disposition along with Chromatograms of DPN metabolites?

6) Revise the conclusion section by including a summary of your key findings, future directions, highlights of your hypothesis as well as improvements compared to already reported work.

7) The reference lists are not updated and sufficient references are not cited in the text.

8) Provide MS spectra with the fragmentation mechanisms of DPN and their metabolites?

6. PLOS authors have the option to publish the peer review history of their article (what does this mean?). If published, this will include your full peer review and any attached files.

Reviewer #1: **Yes**

Reviewer #2: No

---

## [Author Response · Author response to Decision Letter 0]

9 Jun 2023

I have attempted to adjust the manuscript to meet these requirements. Please advise if I have overlooked anything.

I’m not sure what code you are referring to. Is this in reference to the statistical analysis? If that is the case, I can share that.

Could you clarify this for me? Look at the response below. There was no grant for this work. It was a cooperative agreement between the USDA and Bell Laboratories.

This research was not funded through a grant. Funding was provided through a cooperative research agreement between the USDA and Bell Laboratories. Therefore, there is no grant number to provide.

All the residue data generated from this work will be made available.

Reviewer #1 Responses:

Line 30. “The sample was cleaned up with a combination….” Should be stated as “The extracted sample was cleaned up with a combination…”

Correction made.

Line 31. “Separation was achieved….” Should be stated as “chromatographic separation was achieved…”

Correction made.

Line 35 and 37. State precision and accuracy in range instead of average.

Range was added to the average so that both are reported.

Line 73,74, 77, References were cited in form of numbers as they appeared in the document, but some reference were cited with first author name only. Arrange the manuscript as per journal instructions.

Corrections made.

Line 90. “On test day 0 half of the test animals were offered…” should be “On test day one half of the test animals were offered…”

Correction made.

Line 100-102. All animals were euthanized humanely in accordance with American Veterinary Medical Association standards and practices. Cite reference to this statement.

A reference to the AVMA standards (which are available on-line) has been added.

Line 110. DPN is an anticoagulant which is detoxified in the liver. Cite reference to this statement.

A reference for this statement was added.

Line 138-139. ‘The samples were fortified with 1 μg of surrogate dissolved in acetonitrile” should be stated as The samples were spiked with ____µl of surrogate dissolved in acetonitrile (µg/L).

Correction made.

Line 154. accuracy, recovery. Should be stated as accuracy and recovery.

Correction made.

Line 154-155. What does this statement mean “When not specified”, result values were tabulated and analyzed using Microsoft Excel 15.0 (Microsoft Corp, Redmond, WA,USA). This statement seems irrelevant.

This sentence has been reworded to clarify that the bulk of the statistical information (mean, standard deviation, etc) was generated using Excel. The more detailed statistical analysis was performed using SAS.

Line 175. Coefficient of determination was not mentioned. Without coefficient of determination, linearity is meaningless.

This line number is in the methods section of the manuscript; therefore it would not be appropriate to describe the coefficient of determination at this point. The criteria for this parameter is discussed in the results section “Calibration was achieved with a 1/x weighted quadratic regression analysis (Y = aX2 + bX + c) with correlation coefficients of 0.999 or better.” We feel that a discussion of the percent accuracy of the standards is a better representation of fitness of the calibration than correlation, especially over very large calibration curves.

Line 176. Authors are needed to explain how does 0.470 ng/mL to 375 ng/mL is corresponding to a nominal sample concentration of 4.7 ng/g to 3750 ng/g. Authors have mentioned linearity in ng/mL but DL and QL in ng/g. Is there any specific reason to do this? Similarity authors also used positive controls 20, 350 and 12500 ng/g and linearity range 0.470 ng/mL to 375 ng/mL. Authors are needed to remove this ambiguity.

The concentration per volume values (ng/mL) represent the concentration of the final sample extract in the vial as it is injected into the LC/MS/MS. This value is then used to generate the concentration per mass of the liver sample (ng/g). The nominal mass of tissue for each sample was 100 mg. This is extracted into 5 mL of solvent. This results in a dilution factor of approximately 1:50. Therefore, a tissue concentration of 18750 ng/g would produce a response equivalent to 375 ng/mL on the LC/MS/MS. The fortification of controls statement will be made more precise to better reflect this dilution. Calculations could be performed in terms of mass of DPN alone without units of mass or volume, but the industry standard tends to be reporting results in terms of ppb in the tissue (ng/g). A statement making this clearer has been added to the manuscript. Please advise if additional clarification is needed.

Line 204. Authors also required to discuss in detail experiments performed during method development regarding instrumental parameters such mobile phase, gradient elution, ion source parameters etc.

Mobile phase and gradient elution parameters were accepted without modification from previously published works; therefore, no experimental optimization was performed on these parameters. The Masshunter software’s Optimizer function was used to optimize the MRM transitions. A reference to this approach was inserted into the manuscript.

Line 229-232. The DL was defined as the concentration of DPN required to generate a signal equal to 3X the baseline noise (measured peak-to-peak) observed in the baseline at the retention time of DPN in the control samples. The quantitation limit (QL) for DPN in rodent livers was estimated in a similar fashion to the DL with the multiplier 232 of 10X baseline noise used instead of 3X. These lines should be removed as these terms already defined in lines 170-174.

Correction made.

Line 297. a range of <0.1 tp 9.0 μg/g should be written as a range of <0.1 to 9.0 μg/g.

Typographical error was corrected.

Authors did not mention validation guidelines they have used for this study.

I am unsure what this is in reference to? If you mean a particular set of regulatory guidelines, then you are correct. There is no strict regulatory guideline for validation of residue methods of this type. It is a non-food use residue study and does not fall under any specific guideline. The data generated are intended for use in future risk assessments.

Moreover one of the important validation parameter when ESI is used as ion source is ion suppression which was not discussed.

You are correct that no investigation of ion suppression was undertaken as a part of this study. While such a study could be useful, the results of the quality control samples speak for themselves and indicate the performance of the method. The use of a deuterated internal standard also provides a correction for this sort of sample-to-sample variability in method performance and is far more reliable than matrix-matched standards in this case. Furthermore, most of the study samples are of significantly high concentration levels, making such a study less important for this type of data.

Instrumental plotted calibration curve with coefficient of determinations should be shown in the form of figure.

Are you requesting a representative calibration curve, or all the calibration data on a single curve? The study was conducted over several weeks with calibration data generated with each analytical run. The discussion section mentions that all calibration curves were of 0.9999 correlation or better. If the calibration were to fall below this level (and it never did during this study) the run would be rejected and remedial action taken to correct the error. My opinion would be that displaying a straight line has no more value than listing the correlation requirements for the run to be acceptable, but I can add one if required.

Table 1, the gradient program at 5.10 and 5.50 min is same and needed to be corrected.

This is used to indicate the final parameters for the 5.50 minute run time of the gradient. The line will be removed, and a second line added to indicate the final run time.

Reviewer #2 Responses:

1) Abstract, and section headings need to be revised along with more quantitative and background information. The abstract does not summarize the objectives, the principal results, and major conclusions of the present study.

Revision to conform to journal formatting has been completed. I’m unclear what additional quantitative information is required? The results of the study in their entirety are discussed in the manuscript. A revision of the abstract has been performed.

2) The originality of the paper needs to be further clarified. The present form does not have sufficient results to justify the novelty of a high-quality journal paper (there is a lot of similar studies that depicted determination of rodenticide levels). The background of the study, the research gap/scientific novelty, and the state-of-the-art (and purpose) of the study are not clearly illustrated. The hypothesis and justification of the study needs to be elaborated.

The novelty of this research lies in the newly developed bait formulations, not in the analytical method itself. The method is fairly derivative and some modifications to the method presented in the literature have been discussed. As the baits under evaluation are novel and proprietary, a discussion in the literature of their exact formulas and ingredients falls under the umbrella of intellectual property. Additionally, since the baits are a new formulation which has not been widely tested, this is the first report of residue levels in rodents resulting from consumption of the new baits. That is where the novelty of this research lies. Efforts have been made to emphasize this in the manuscript.

3) Have you looked for the chirality of the DPN/and their stereoisomers through chiral column analysis? What about the optimization of the sample preparation process and LC-MS/MS conditions?

Investigation of the isomers of DPN has not been conducted as a part of this study. There is no peak splitting to indicate the separation of isomers using the present analytical technique. As the goal of this research was to quantify the concentration of parent (regardless of any isomeric form) in the livers, examination of isomers was not undertaken.

The sample preparation and LC/MS/MS conditions were investigated as discussed in the manuscript. Previous methods developed in our facility optimized the LC conditions (mobile phase, stationary phase, column flow, etc) for the separation of rodenticides in biological matrices. Therefore, no additional time was spent on these experiments. The primary focus of optimization was in the area of sample extraction and clean-up as discussed in the manuscript.

4) Any information about matrix-matched standards and fortified liver matrix sample preparation and evaluation of the extraction procedures?

Matrix-matched standards are often not possible or feasible for wild-caught animals. Biological and dietary variation for the population is much more significant than in lab strains used for clinical research. In addition, the limited amount of sample material prevented the preparation of matrix-matched standards. A deuterated internal standard was used, which should correct for any matrix variability in the samples.

All the fortified quality control samples detailed in the manuscript were prepared in control liver tissues. This is the primary way in which method performance was evaluated and is reflected in the reported results.

5) What about metabolites and their disposition along with Chromatograms of DPN metabolites?

Metabolism of DPN does occur to a small degree, but there are no commercially available standards for the quantification of such metabolites. Therefore, no quantitative results could be generated. Qualitative results of the presence of metabolites would not be particularly useful as DPN is known to metabolize via the CYP 450 pathways of the liver. Additionally, for comparison to literature values of DPN residues in other studies the metabolites would not be useful as other studies of the DPN residues from field applications do not report the metabolites.

6) Revise the conclusion section by including a summary of your key findings, future directions, highlights of your hypothesis as well as improvements compared to already reported work.

A revision of the conclusion section has been performed. Future directions for this work are not being considered at this time as the research objectives have been achieved. The method was developed and validated and the residue levels in the study animals were reported. We are not trying to improve on already-reported methods in the literature. We do highlight changes to the methods we used to improve performance for the research discussed in the manuscript.

7) The reference lists are not updated and sufficient references are not cited in the text.

The reference list was modified to fit journal standards. I’m not sure what additional references would contribute to the manuscript. How many are required to meet this request? We feel sufficient references have been cited to describe the foundation for the research undertaken.

8) Provide MS spectra with the fragmentation mechanisms of DPN and their metabolites?

MS spectra are usually not very instructive in an MRM method paper as it is a targeted assay. However, one can be generated to meet this request.

Metabolites were not the focus of this research and no data on the metabolites of DPN have been collected.

---

## [Decision Letter · Decision Letter 1]

17 Jul 2023

Determination of residue levels of rodenticide in rodent livers offered novel diphacinone baits by liquid chromatography-tandem mass spectrometry

PONE-D-23-07599R1

Dear Dr. Goldade,

We’re pleased to inform you that your manuscript has been judged scientifically suitable for publication and will be formally accepted for publication once it meets all outstanding technical requirements.

Kind regards,

Totan Adak

Academic Editor

PLOS ONE

Additional Editor Comments (optional):

Reviewers' comments:

Reviewer's Responses to Questions

**Comments to the Author**

1. If the authors have adequately addressed your comments raised in a previous round of review and you feel that this manuscript is now acceptable for publication, you may indicate that here to bypass the “Comments to the Author” section, enter your conflict of interest statement in the “Confidential to Editor” section, and submit your "Accept" recommendation.

Reviewer #1: All comments have been addressed

Reviewer #2: All comments have been addressed

2. Is the manuscript technically sound, and do the data support the conclusions?

Reviewer #1: Yes

Reviewer #2: Partly

3. Has the statistical analysis been performed appropriately and rigorously? 

Reviewer #1: Yes

Reviewer #2: Yes

4. Have the authors made all data underlying the findings in their manuscript fully available?

Reviewer #1: Yes

Reviewer #2: Yes

5. Is the manuscript presented in an intelligible fashion and written in standard English?

Reviewer #1: Yes

Reviewer #2: Yes

6. Review Comments to the Author

Reviewer #1: The manuscript has been largely revised by Authors and have addressed all of my concerns. Hence got my recommendation for publication.

Reviewer #2: Although authors have provided to some extent justifications for the raised queries still the reviewer felt the following information is missing:

a) Provide MS spectra with the fragmentation mechanisms of DPN and their

metabolites?- Justifications are not clear

b) Toxicity summary of the DPN needs to be addressed properly with the heat map

Above all the responses have been given in a non-polite way which is not acceptable!!!

7. PLOS authors have the option to publish the peer review history of their article (what does this mean?). If published, this will include your full peer review and any attached files.

Reviewer #1: **Yes: **Muhammad IMRAN (IMRANFSTOX@GMAIL.COM)

Reviewer #2: No

---

## [Editor Report · Acceptance letter]

27 Jul 2023

PONE-D-23-07599R1 

Determination of residue levels of rodenticide in rodent livers offered novel diphacinone baits by liquid chromatography-tandem mass spectrometry 

Dear Dr. Goldade:

I'm pleased to inform you that your manuscript has been deemed suitable for publication in PLOS ONE. Congratulations! Your manuscript is now with our production department. 

Kind regards, 

on behalf of

Dr. Totan Adak 

Academic Editor

PLOS ONE